# VISGate: ROI-Conditioned Dual-Head Encoders that Align Visual Features and Brain Responses

**Morteza Mahdiani**[*]
Mila - Québec AI Institute
Université de Montréal

**Ian Charest**
Mila - Québec AI Institute
Université de Montréal

## Abstract

Foundation models enable image-to-brain encoders that scale across cortical regions and subjects. **VISGate** couples a frozen DINOv2 backbone with a lightweight ROI-Transformer and two outputs: (i) *voxel-wise fMRI response prediction* and (ii) *per-ROI caption embedding prediction*. Trained on the Natural Scenes Dataset (NSD), the model yields robust voxel predictivity, with systematic variations across five cortical streams (early, midventral, midlateral, ventral, lateral). We evaluate per-voxel correlation, split-half noise ceilings, and normalized accuracy, and we visualize semantic category–wise ROI profiles. Across multiple NSD subjects, ventral and lateral ROIs dominate normalized accuracy, while the caption heads emphasis early and lateral ROIs, suggesting distinct and shared contributions of visual and linguistic components in brain responses to natural scenes.

## 1 Introduction

Image-to-brain encoding probes how visual features align with human cortex. Prior NSD encoders typically optimized either *voxel prediction* or *text alignment*, seldom both jointly within one pipeline, even though practical systems increasingly benefit from models that are simultaneously brain-predictive and text-grounded. Brain-predictive models ensure faithful neural modeling, while text grounding enables user-facing tasks (retrieval, prompting, reporting) and supports zero-/few-shot generalization. We target a compact encoder with a frozen DINOv2 backbone and ROI-wise shared representations that expose two interfaces: one for *voxel-wise, ROI-conditioned prediction* and one for *per-ROI caption-embedding alignment*.

Two main questions arise: (Q1) Where do neural predictivity and semantic alignment *coincide* versus *diverge* across visual ROIs on NSD? (Q2) How robust are these effects across subjects when using common `fsaverage` ROI masks [2, 5] and a *shared* backbone and ROI decoder?

We address these with a single-layer ROI-Transformer that emits five ROI tokens feeding dual heads (voxels, captions), and with a joint loss that combines *per-voxel mean squared error (MSE) between predicted and measured fMRI responses* with *per-ROI text-embedding alignment* (cosine + MSE), weighted by $\lambda_{\cos}, \lambda_{\mathrm{emb}}, \lambda_{\mathrm{vox}}$. We also report a *bounded* normalized accuracy alongside raw $r$ and noise ceilings estimated via split-half reliability with 100 Monte Carlo splits [15, 2].

**Contributions.**

- A compact ROI-Transformer atop a frozen DINOv2 with dual heads (voxel, caption).

- A joint objective that reveals distinct and shared contributions between neural predictivity and semantic grounding.

---

[*]Correspondence to m72.morteza@gmail.com.

39th Conference on Neural Information Processing Systems (NeurIPS 2025) Workshop: UniReps.

Overall, VISGate builds on top of the Doerig et al. [5] results that showed high-level visual cortex aligned with LLM-derived caption semantics. We adopt the same conceptual stance, and ask the following question: where and how is semantic information expressed across the cortex? VISGate uses a shared latent visual representation with task-specific readouts (voxel and caption heads), so any differences reflect readout specialization rather than disjoint representational spaces. With ROI-conditioning, we localize and interpret the division of labor across areas, showing convergence with caption space in high-level ROIs while adding ROI-level interpretability. Together, the text-centric and ROI-aware visual approaches offer additive coverage and a more complete account of brain–AI alignment.

## 2 Related Work

**Voxel prediction encoders.** Early NSD models coupled CNN features with receptive-field weighting or ROI-wise readouts, and later work introduced explicitly brain-optimized networks such as GNet and its successors [18, 15]. More recent approaches have explored pruning and compact transformer architectures for efficient voxel prediction on NSD, including BOLDreams [7] and Brainformer [13], which improve efficiency while maintaining accuracy. These models, however, are primarily optimized for voxel prediction and do not explicitly incorporate text-based objectives.

**Cross-modal alignment.** Parallel work emphasizes alignment between vision and language. CLIP [16] and MPNet [17] provide large-scale text embeddings widely used for retrieval, motivating brain encoders that leverage shared visual–linguistic spaces. Recent studies show that human high-level visual representations align with language models [5], motivating multimodal objectives in encoding. Adeli et al. [1] further demonstrate that transformer-based brain encoders with attention routing can improve predictivity and interpretability. VISGate extends this line by using a shared visual latent that jointly optimizes voxel prediction and per-ROI caption-embedding alignment.

**Cross-subject and personalization.** Beyond single-subject training, Allen et al. introduced the NSD dataset to facilitate large-scale comparisons across subjects [2]. Personalized encoders trained on smaller per-subject datasets achieve robust performance [6], while universal encoders trained jointly across multiple participants demonstrate improved generalization [3]. Our work builds on this progression by designing a dual-head, ROI-conditioned encoder that unifies voxel prediction and semantic alignment in a compact architecture.

## 3 Method

**Subject-consistent ROI masks and cross-subject transfer.** All subjects are mapped to the `fsaverage` cortical surface; therefore, ROI masks are defined in a *common* surface space with consistent vertex indices and topology across subjects. In our setting, the five visual streams are specified as binary masks on `fsaverage`, and the same per-ROI vertex sets are used for every subject. Consequently, we do not require subject-specific voxel (vertex) heads: a *single* linear map per ROI operates on a fixed vertex index set shared across subjects, while the backbone (DINOv2) [14] and the ROI decoder are also shared. This design avoids any ad hoc padding or remapping across subjects and allows direct cross-subject evaluation with common ROI geometry. However, alignment to `fsaverage` only approximates functional correspondence: voxels that map to the same surface vertex across subjects need not share identical tuning, so the shared per-ROI heads should be viewed as imposing a shared functional prior rather than assuming perfect homology. We visualize the five stream masks on `fsaverage` (App. Fig. 3) and report the per-ROI vertex counts (App. Table 2).

**Backbone and ROI decoder.** We freeze a DINOv2 backbone to extract image tokens $\{x_i\}$. A single-layer Transformer decoder (ROI-Transformer) with $n_{\mathrm{ROI}} = 5$ learned queries attends to the backbone tokens and produces ROI embeddings $\{z_r\}$ [18, 9].

**Dual heads.** *Voxel head:* each ROI token $z_r$ is mapped to its voxel slice via a linear head, yielding predictions $\hat{\mathbf{y}}_r$ for the vertices in that ROI. *Caption head:* ROI-specific MLPs project $z_r$ into a fixed text-embedding space (MPNet) [17], producing predicted caption embeddings $\hat{\mathbf{e}}_r$.

**Loss.** For voxels $\mathbf{y}$ and per-ROI text embeddings $\mathbf{e}$, predictions $\hat{\mathbf{y}}$ and $\hat{\mathbf{e}}_r$ are optimized with

$$\mathcal{L} = \lambda_{\mathrm{cos}}\big(1 - \cos\langle \hat{\mathbf{e}}_r, \mathbf{e}\rangle\big) + \lambda_{\mathrm{emb}}\big\|\hat{\mathbf{e}}_r - \mathbf{e}\big\|_2^2 + \lambda_{\mathrm{vox}}\big\|\hat{\mathbf{y}} - \mathbf{y}\big\|_2^2.$$

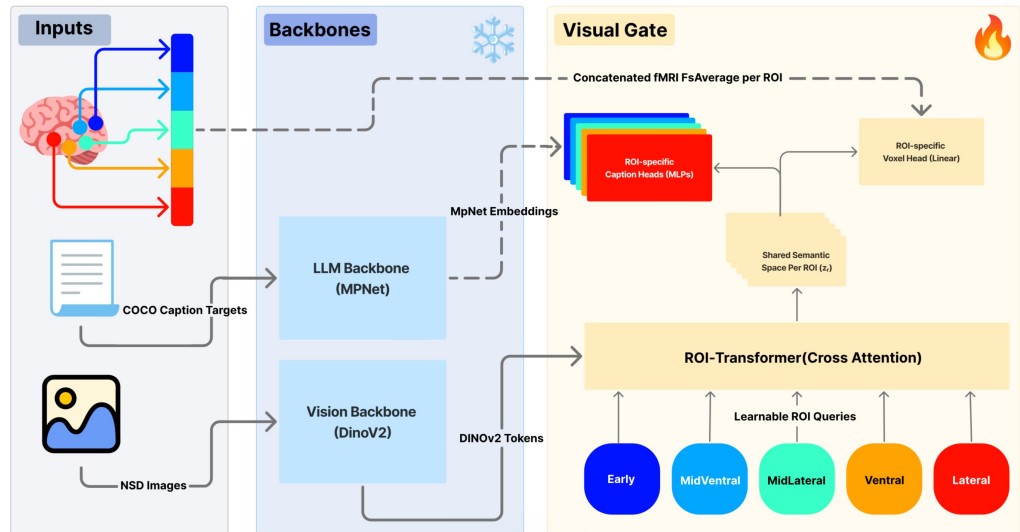

Figure 1: **VISGate architecture.** An input image is encoded into tokens by a frozen DINOv2 backbone. A lightweight ROI-Transformer decoder generates ROI-specific embeddings (early, midventral, midlateral, ventral, lateral). Two parallel heads operate on these ROI tokens: (i) a caption head that projects embeddings into a fixed text-embedding space (MPNet) and is trained with cosine and MSE losses, and (ii) a voxel head that predicts voxel activations for each ROI using MSE against fMRI betas. This dual-head design enables simultaneous optimization for semantic alignment and voxel predictivity.

**Training.** We implement VISGate in PyTorch, using Adam (learning rate $10^{-4}$), batch size 32, dropout 0.5, and training for 20 epochs with an 80/20 train–validation split. Model selection is based on lowest validation loss. Full hyperparameter and environment details are provided in Appendix C.

## 4 Evaluation Protocol

**Data.** We train VISGate on approximately 9k image–response pairs from subject 01 of NSD. Evaluation is performed on the standardized `shared1k` image set. For voxel prediction, we test on subjects 01, 02, 05, and 07, yielding both within-subject performance (subject 01) and cross-subject generalization (02, 05, 07) on a shared stimulus set. For caption retrieval, evaluation is image-based and subject-agnostic: given a fixed backbone and ROI-Transformer, the caption heads produce identical outputs for a given image regardless of subject, so caption metrics are computed once over the `shared1k` gallery.

**Voxel metrics.** For each voxel $v$, we compute the Pearson correlation $r_v$ between predicted and measured responses on held-out test images, following established NSD evaluation protocols [2]. To account for measurement reliability, noise ceilings $\mathrm{NC}_v$ are estimated via split-half reliability across stimulus repeats and corrected with the Spearman–Brown formula [15]. We then define a bounded, ceiling-normalized accuracy:

> **Normalized accuracy (bounded).** For voxel $v$,
>
> $$\mathrm{Acc}_v \; = \; \min\!\Big(1, \, \frac{r_v^2}{\max(\varepsilon, \, \mathrm{NC}_v)}\Big) \; = \; \frac{\min\!\big(r_v^2, \, \max(\varepsilon, \, \mathrm{NC}_v)\big)}{\max(\varepsilon, \, \mathrm{NC}_v)}, \qquad \varepsilon = 10^{-8}.$$
>
> This guarantees $0 \leq \mathrm{Acc}_v \leq 1$, with $\mathrm{Acc}_v = 1$ when $r_v^2 \geq \max(\varepsilon, \, \mathrm{NC}_v)$ (and in particular when $r_v^2 \geq \mathrm{NC}_v$ if $\mathrm{NC}_v \geq \varepsilon$).

Table 1: Normalized Accuracy ($r^2/NC$) by ROI and Subject on NSD shared1k.

| Subject | Early | Mid-Ventral | Mid-lateral | Ventral | Lateral |
|---------|-------|-------------|-------------|---------|---------|
| 01 | 0.32 | 0.45 | 0.41 | **0.58** | 0.55 |
| 02 | 0.28 | 0.42 | 0.38 | **0.53** | 0.50 |
| 05 | 0.30 | 0.44 | 0.40 | **0.56** | 0.52 |
| 07 | 0.29 | 0.43 | 0.39 | **0.54** | 0.51 |

Thus, $\text{Acc}_v$ expresses the fraction of explainable variance captured by the model [2]: $\text{Acc}_v = 1.0$ indicates performance at the estimated noise ceiling, while lower values reflect underperformance relative to voxel reliability.

**Caption metrics.** For the caption head, ROI-wise predicted embeddings $\hat{\mathbf{e}}_r$ are compared to ground-truth caption embeddings using cosine similarity on the shared1k images. Retrieval performance is quantified with standard metrics over the 1k-image gallery, including mean reciprocal rank (MRR) and recall at $K$ (R@1/5/10) [12, 16]. See Appendix Table 3 for full metric definitions and additional results. These caption metrics provide a complementary view of semantic alignment alongside voxel-level predictivity, but they are independent of subject because the caption head does not take subject-specific information as input.

## 5 Results

**Cross-subject transfer.** Training on subject 01 transfers well to subjects 02, 05, and 07. While the raw voxel correlations diminish slightly, the ROI patterns remain robust: ventral and lateral ROIs consistently lead in normalized accuracy (see Table 1).

These results confirm the dominance of ventral cortex in normalized accuracy across subjects, with lateral closely following, while early and mid-level ROIs lag behind. This pattern is consistent across participants and suggests that idiosyncratic tuning in early visual cortex converges onto more stable high-level response distributions in ventral and lateral streams, supporting cross-subject generalization for natural scenes.

**Caption and voxel profiles across semantics.** Figure 2 shows category-wise ROI profiles, where performance is z-scored within each semantic group to highlight *relative* ROI contributions (ranking within category) rather than absolute performance. Panel 2a plots z-MRR for the caption head, and Panel 2b plots z-normalized accuracy for the voxel head.

Two patterns emerge. First, both heads share a similar *semantic fingerprint*: inanimate, indoor, tool/non-food, and big/small categories tend to be easier overall than animals and outdoor scenes, in line with classic high-level ventral tuning for object and scene categories. Second, once we z-score within category, the ROIs that rank highest can differ between heads. For the caption head, lateral cortex is consistently near the top across categories, with early visual cortex sometimes rising above mid-level and ventral ROIs. For the voxel head, ventral cortex dominates the z-normalized accuracy profiles, with lateral typically second and early reliably lowest. This yields complementary ROI specializations: captions draw more heavily on lateral (and occasionally early) ROI tokens, whereas voxel prediction relies most strongly on ventral and lateral tokens.

We quantified this by correlating z-MRR and z-normalized accuracy across categories within each ROI. These correlations are generally weak and can even be negative in some ROIs, despite strong positive correlations between the *raw* caption and voxel metrics. This is exactly what the z-scoring does: at the absolute level the two heads tend to rise and fall together (shared category difficulty and SNR), but after removing per-category magnitude, the analysis asks a different question: *which ROIs rank higher than others for this category?* Under that lens, the caption and voxel heads do not always highlight the same ROIs and can even anti-rank, suggesting a division of labor in how different cortical streams support semantic readout versus voxel predictivity.

**Relation to prior work.** These findings resonate with (and refine) prior observations. Doerig et al. [5] showed that high-level visual cortex expresses category and semantic structure that is well captured by LLM-derived caption embeddings, beyond what would be expected from simple low-level visual features. Our voxel results, where ventral and lateral ROIs dominate normalized

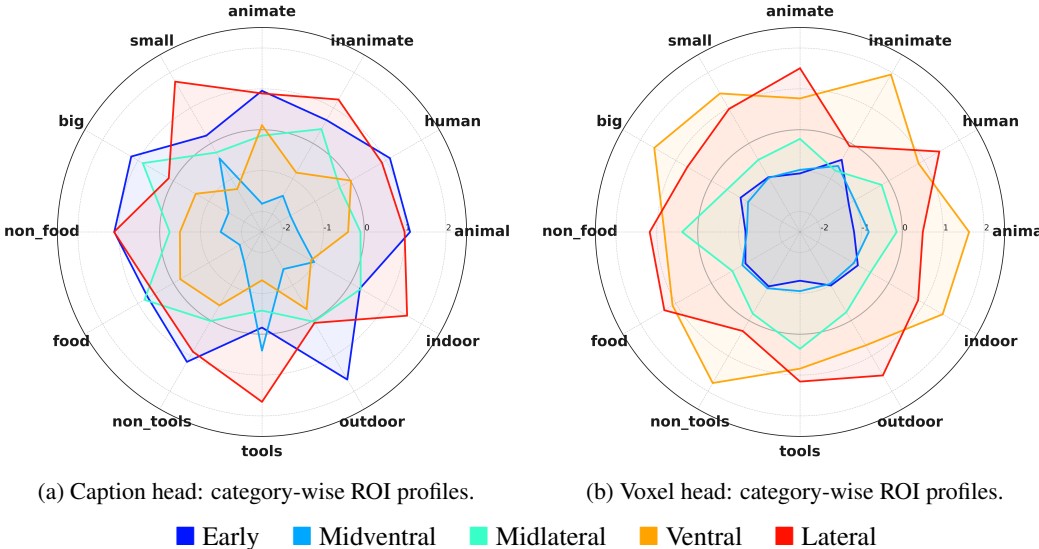

(a) Caption head: category-wise ROI profiles.    (b) Voxel head: category-wise ROI profiles.

■ Early   ■ Midventral   ■ Midlateral   ■ Ventral   ■ Lateral

Figure 2: **Caption and voxel profiles across semantics.** Each curve is a single ROI (early, midventral, midlateral, ventral, lateral). Panel a shows z-scored mean reciprocal rank (z-MRR) for the caption head, and Panel b shows z-scored normalized accuracy for the voxel head, using ROI-wise averages of normalized accuracy aggregated over voxels for each semantic group. Values are z-scored within each semantic group (across ROIs), so peaks reflect *relative* ROI contributions for each category rather than absolute performance.

accuracy, are consistent with that picture. Adeli et al. [1] further demonstrated that transformer-based brain encoders with attention routing to category-selective areas can improve voxel prediction and interpretability. VISGate extends this line of work by placing semantic alignment and voxel prediction into a single dual-head framework and by directly quantifying how their ROI-wise profiles converge and diverge across categories, highlighting complementary specializations rather than a single unified "best" ROI for all objectives.

## 6  Limitations and Outlook

**Limitations.** (1) Our ROI set is deliberately coarse; finer parcellations or subject-adaptive ROI queries may better capture functionally specific subregions. (2) All subjects are aligned to `fsaverage`, which provides a common surface geometry but only an approximate functional correspondence; voxels mapped to the same vertex need not share identical tuning, so our shared per-ROI heads should be interpreted as imposing a shared prior rather than assuming perfect homology. (3) While the caption head provides text-grounded interpretability, stronger contrastive curricula and negative sampling could further disentangle overlapping semantics. (4) Cross-subject generalization, though promising, remains constrained; adapters or hypernetworks may improve robustness. (5) We evaluate two objectives only, which limits the breadth of behavioral and cross-modal conclusions.

**Scope.** Voxel-wise prediction (neural predictivity) and per-ROI caption-embedding alignment (semantic grounding) are *representative* probes of (i) how well a model explains brain responses and (ii) how those responses map into task language, but they are not exhaustive. Broader task families include behaviorally anchored objectives (e.g., recognition accuracy, reaction time), cross-modal grounding (audio, language-only prompts), and temporally resolved signals (ECoG/MEG), each potentially emphasizing different computations.

**Outlook.** We see three practical extensions: (i) multi-head training that adds object/attribute classifiers, category logits, cross-modal retrieval, or generative decoders; (ii) multi-subject training with parameter-efficient adapters and lightweight per-subject calibration atop shared `fsaverage` geometry; and (iii) biologically informed region hierarchies and closed-loop paradigms (stimulus selection under ROI constraints). These directions will test whether the complementary ROI patterns we observe persist beyond the two objectives studied here or fragment along task and timescale.

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

# A ROI details

## A.1 Region Masks Used for ROI Conditioning

**Provenance.** Masks are taken directly from the NSD visual-stream parcellations on `fsaverage` and provided by the NSD authors; we do not modify these region definitions [2, 5].

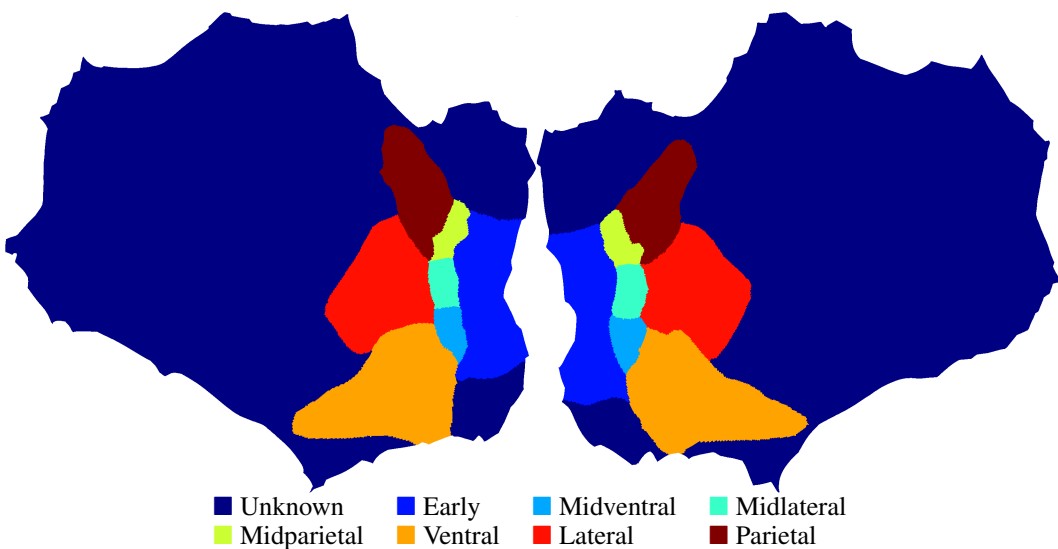

Figure 3: NSD visual-stream masks on `fsaverage` (flattened). Labels: Unknown, Early, Midventral, Midlateral, Midparietal, Ventral, Lateral, Parietal. Masks are defined in a common `fsaverage` surface (consistent vertex indices across subjects). We analyze the following visual-pathway ROIs: **Early**, **Midventral**, **Midlateral**, **Ventral**, and **Lateral**; the remaining labels are shown for completeness only.

## A.2 Per-ROI voxel counts

Table 2: Voxel (surface-vertex) counts on `fsaverage` for visual-pathway ROIs (left+right hemispheres, common masks). **Total across ROIs: 55,446 vertices**.

| ROI | Number of vertices |
|---|---|
| Early | 11,389 |
| Midventral | 1916 |
| Midlateral | 2286 |
| Ventral | 19,065 |
| Lateral | 20,790 |
| **Total** | **55446** |

## A.3 Empirical Voxel-Space Similarity on Shared1k (RSA/CKA)

We compute cross-subject voxel-space similarity on the `shared1k` image set using the *measured* fMRI responses (betas), not model predictions. For each subject and ROI, we project betas to the common `fsaverage` surface and form stimulus–stimulus RDMs and voxel-pattern matrices from the corresponding shared1k betas.

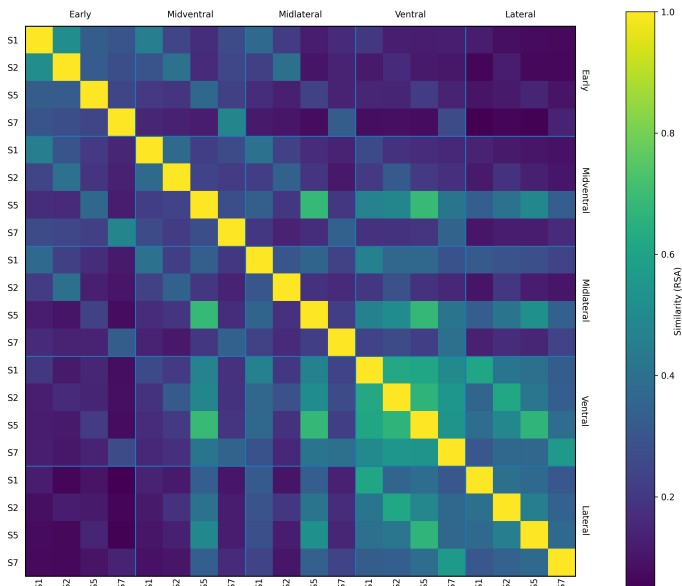

Figure 4: **Cross-subject RSA across ROIs.** Heatmap of representational similarity analysis (RSA; [11]) over all pairs of subjects and ROIs, computed on measured `fsaverage`-space fMRI betas for the `shared1k` image set. The figure uses hierarchical axis labels: major labels (top/left) denote regions of interest (ROIs); minor labels within each ROI block denote subject IDs. Each cell is the Pearson correlation between upper triangles of stimulus–stimulus RDMs ($1 - \mathrm{corr}$ across voxels). Warmer colors indicate greater agreement in representational geometry.

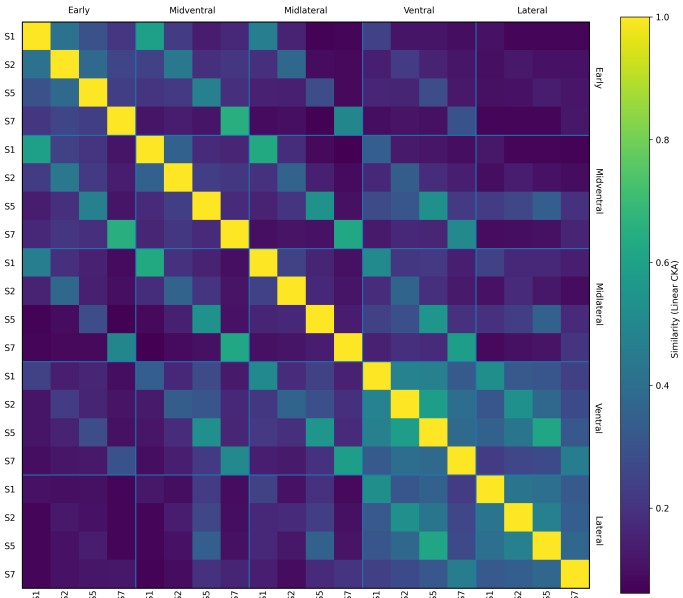

Figure 5: **Cross-subject linear CKA across ROIs.** Same layout as Fig. 4, but entries are linear centered kernel alignment (CKA; [10]) between stimulus-by-voxel patterns derived from measured `fsaverage`-space `shared1k` fMRI betas. CKA captures pattern-level alignment that is invariant to isotropic scaling, complementing RSA's geometry-based comparison.

# B  Category-wise retrieval metrics

## B.1  Retrieval metrics: raw values

Following standard image–text retrieval practice, we report Recall@K and Median Rank [8] and include Mean Reciprocal Rank [4] for rank-quality summarization.

Table 3: MRR per ROI (Early, Midventral, Midlateral, Ventral, and Lateral) and overall macro (unweighted average) metrics (R@1, R@5, R@10, MRR, MedR) for the reported canonical categories.

| Group | ROI (MRR) | | | | | Overall (macro across ROIs) | | | | |
|---|---|---|---|---|---|---|---|---|---|---|
| | Early | Midventral | Midlateral | Ventral | Lateral | R@1 | R@5 | R@10 | MRR | MedR |
| animate | 0.216 | 0.196 | 0.208 | 0.210 | 0.215 | 0.086 | 0.317 | 0.491 | 0.209 | 11.00 |
| inanimate | 0.207 | 0.195 | 0.206 | 0.199 | 0.211 | 0.079 | 0.306 | 0.500 | 0.203 | 10.60 |
| human | 0.238 | 0.224 | 0.231 | 0.232 | 0.237 | 0.105 | 0.348 | 0.522 | 0.233 | 9.60 |
| animal | 0.245 | 0.221 | 0.234 | 0.231 | 0.243 | 0.104 | 0.353 | 0.535 | 0.239 | 9.40 |
| indoor | 0.229 | 0.216 | 0.230 | 0.215 | 0.243 | 0.104 | 0.325 | 0.497 | 0.227 | 10.20 |
| outdoor | 0.224 | 0.213 | 0.218 | 0.217 | 0.218 | 0.095 | 0.320 | 0.492 | 0.219 | 11.00 |
| tools | 0.195 | 0.199 | 0.193 | 0.188 | 0.207 | 0.084 | 0.285 | 0.463 | 0.196 | 12.20 |
| non_tools | 0.209 | 0.184 | 0.198 | 0.195 | 0.206 | 0.080 | 0.297 | 0.467 | 0.199 | 11.80 |
| food | 0.155 | 0.129 | 0.157 | 0.146 | 0.154 | 0.064 | 0.196 | 0.328 | 0.154 | 23.40 |
| non_food | 0.193 | 0.175 | 0.184 | 0.182 | 0.193 | 0.071 | 0.276 | 0.437 | 0.185 | 13.00 |
| big | 0.239 | 0.215 | 0.236 | 0.223 | 0.230 | 0.099 | 0.340 | 0.531 | 0.229 | 9.70 |
| small | 0.244 | 0.236 | 0.238 | 0.226 | 0.262 | 0.112 | 0.348 | 0.566 | 0.241 | 8.80 |

## B.2  Broad (overlapping) COCO groups

Table 4 summarizes the main image-level groups used in our analyses.

| Group | Type | Definition (image-level) |
|---|---|---|
| animate | broad | At least one object with category name `person` or supercategory `animal`. |
| inanimate | broad | No `person` and no `animal` supercategory present. |
| human | broad | At least one object with category name `person`. |
| animal | broad | At least one object whose supercategory is `animal`. |
| tools | broad | At least one object from a predefined tool list (e.g., `knife`, `fork`, `remote`, `tennis racket`, …). |
| non_tools | complement | No object from the tool list present. |
| food | broad | At least one object from a predefined food list (e.g., `banana`, `pizza`, `cake`, …). |
| non_food | complement | No object from the food list present. |
| big | broad | At least one object from a list of large items (e.g., `car`, `bus`, `sofa`, `bed`, …). |
| small | broad | At least one object from a list of small items (e.g., `cell phone`, `mouse`, `book`, …). |
| indoor | broad | At least one object typically found indoors (e.g., `bed`, `microwave`, `refrigerator`, …). |
| outdoor | broad | At least one object typically found outdoors (e.g., `bicycle`, `car`, `skis`, `surfboard`, …). |

Table 4: Broad, overlapping COCO-derived image groups. An image may belong to several groups simultaneously (e.g., both `human`, `tools`, and `indoor`).

## C  Model and Training Details

**Objective.**  We optimize a joint loss over images and ROIs:

$$\mathcal{L} = \lambda_{\text{vox}} \, \text{MSE}(\hat{\mathbf{y}}, \mathbf{y}) \; + \; \lambda_{\text{cos}} \left( 1 - \cos(\hat{\mathbf{e}}, \mathbf{e}) \right) \; + \; \lambda_{\text{emb}} \, \text{MSE}(\hat{\mathbf{e}}, \mathbf{e}),$$

with $\lambda_{\text{cos}}{=}0.7$, $\lambda_{\text{emb}}{=}0.3$, and $\lambda_{\text{vox}}{=}0.7$ (this run). Here $\hat{\mathbf{y}}$ are predicted voxel responses, $\mathbf{y}$ are fMRI betas, $\hat{\mathbf{e}}$ are predicted per-ROI text embeddings, and $\mathbf{e}$ are MPNet embeddings.

### C.1  Hyperparameters and Environment

Table 5: System, software, and training configuration.

| Parameter | Value |
|---|---|
| Backbone | Frozen DINOv2 |
| ROI decoder | Single layer, $n_{\text{heads}}{=}8$, FF dim $= 4D$ (1024) |
| Caption head loss weights | $\lambda_{\text{cos}}{=}0.7$, $\lambda_{\text{emb}}{=}0.3$ |
| Voxel head loss weight | $\lambda_{\text{vox}}{=}0.7$ |
| Optimizer | Adam (default $\beta$) |
| Learning rate | $1.0 \times 10^{-4}$ |
| Weight decay | 0.03 |
| Dropout | 0.5 |
| Batch size | 32 |
| Epochs | 20 |
| Seed | 314 |
| Subject / split | subj01 / `train9k` |
| OS | Linux 4.18 (`x86_64`, glibc 2.30) |
| GPU | $1 \times$ NVIDIA A100-SXM4-40GB |
| CPU (logical) | 64 |
| Python | CPython 3.10.2 |
| Key libs | HuggingFace 4.36.2, W&B 0.19.9 |

### C.2  Model parameters and core components

Table 6: **VISGate model architecture (core components)**.

| Component | Details |
|---|---|
| Backbone | DINOv2 (base), frozen |
| Hidden size ($D$) | 768 |
| ROIs ($R$) | 5 (from `roi_bounds`) |
| Queries | $R \times D$ learnable parameters |
| Decoder | TransformerDecoder (1 layer, $n\_head = 8$, FF $= 4D$, dropout 0.3) |
| Voxel heads | $R$ linear layers ($D \rightarrow V_i$), no bias |
| Caption heads | $R$ MLPs: $D \rightarrow 1024 \rightarrow 768$ |
| Trainable params | Queries + decoder + heads (backbone frozen) |

Table 7: **Model size and compute.** Parameter counts include backbone; trainable excludes the frozen DINOv2. MACs reported per forward pass; we use the convention $1\,\text{MAC} \approx 2\,\text{FLOPs}$.

| Quantity | Value |
|---|---|
| Total parameters | 387,141,376 |
| Trainable parameters | 82,772,736 |
| Frozen parameters | 304,368,640 |
| Estimated compute (MACs) | 10.17 G-MACs / forward |
| Estimated compute (FLOPs) | $\approx 20.34$ GFLOPs / forward |

## C.3 Computational Complexity: Big-$\mathcal{O}$ Notation

| Symbol | Meaning |
|--------|---------|
| $B$ | Batch size |
| $R$ | Number of ROIs (queries) |
| $S$ | Backbone token length (incl. CLS) |
| $D$ | Hidden size (decoder/heads width) |
| $V$ | Total voxels ($V = \sum_{i=1}^{R} V_i$) |

**Notation.**

**Per-forward cost (one decoder layer).** For a Transformer decoder with $R$ query tokens attending to $S$ backbone tokens and per-ROI linear heads:

$$\text{Self-attn (queries)}: \ \mathcal{O}\big(B(R^2 D + RD^2)\big)$$
$$\text{Cross-attn (q×mem)}: \ \mathcal{O}\big(B(RSD + (R+S)D^2)\big)$$
$$\text{FFN (width} \approx 4D): \ \mathcal{O}\big(BRD^2\big)$$
$$\text{Voxel heads } (D \to V_i): \ \mathcal{O}\big(BVD\big)$$

**Overall complexity.** Aggregating dominant terms for one forward pass:

$$\boxed{\mathcal{O}\big(B\,[\,VD \ + \ RSD \ + \ (R+S)D^2\,]\big)}$$

In NSD-like settings where $V$ is large (e.g., $\sim 5.5 \times 10^4$), the voxel-head term $BDV$ typically dominates wall-clock cost.

