# OpenReview forum: "VISGate: ROI-Conditioned Dual-Head Encoders that Align Visual Features and Brain Responses"
_NeurIPS.cc/2025/Workshop/UniReps — UniReps2025_

### Official Review · Reviewer_XbUa · 2025-09-11
**ROI-conditioned dual encoder architecture design reveals interesting alignment findings for visual features and brain responses. Some details and discussions are lacking. Weak Accept**

**Confidence:** 3

**Review:**

Summary:
This work proposes a dual-head encoder architecture conditioned on ROIs that is capable of two downstream tasks: 1) predict per ROI caption embeddings (text grounded); 2) regress voxel activations (brain predictive). The architecture comprises a DINOv2 model coupled with an ROI transformer models,connected to encoder heads for predictions on target functionalities. Evaluations on NSD dataset demonstrate that the voxel encoder achieves best performance at the ventral region and generalizes across subjects, while the caption prediction encoder results demonstrate divergence from the voxel predictions based on the image category, which is an interesting finding justifying the need for dual-head architectures as per the authors claims.

Overall Review
I believe the work suited for the workshop considering their alignment findings. I would have liked to understand a bit more the broader umbrella under whic ROI-conditioned downstream tasks - voxel and caption predictions -- fall to understand if they are representative enough of their broader categories. Otherwise, experiments with different form of downstream tasks are needed to corroborate conclusions on alignment. If the latter is true, then that could even justify a 'multi-head' architecture, and a discussion on whether that is viable or limiting would have provided better context. Some details are also lacking especially regarding the justification for architecture design and loss functions. But overall, I think the alignment findings can be valuable for the workshop though I hope some of the points raised are better clarified.

Strengths:
- Interesting findings on the alignment and generalization performance across subjects for voxel predictions
- Good explanation on the metrics and experimental setup
- Paper is to the point and relevant to UniReps theme

Weakness
- More details on the broader categories of the ROI-based downstream tasks. If the ones proposed are sufficient to draw the conclusions, then kindly explain
- Motivation for dual head architecture design instead of having multiple individual architectures each geared towards a certain task. Some discussion here would be appreciated
- Details for architectural choices (MLP of encoders dimensions and so)
- loss function choices: e.g., why is MSE needed on top of cosine similarity for alignment embeddings? It is not clear to me why two terms for alignment are needed

Quality
Paper has proper quality, introduces the setting and states its contributions clearly, before discussing related work in the field, and describing their method. Experiments reveal interesting findings regarding the stated problem

Clarity
Writing is clear. Font sizes in the figures are difficult to read at times

Originality
The architecture design seems original. Though it is not clear whether the alignments findings were discovered in previous works involving separate models for the different downstream tasks.

Significance
I find the alignment findings fairly significant. It motivates the need to have both text-base grounding and brain predictive tasks to understand the relations between ROIs and different stimuli categories.

Comments:
- Lines 15 emphasis on why practical system require both is needed.
- Figure 1 font is difficult to read
- lines 36 efficient voxel prediction. As that was mentioned, can we understand when efficiency is relevant in this context? If it is then there is also a missing discussion on the efficiency of the dual-head encoder design

**Score:**

3

**Topic Fit:**

3

---

### Official Review · Reviewer_YoWX · 2025-09-14
**Promising ROI-conditioned dual-head; fix metric bug and add baselines (Weak Accept)**

**Confidence:** 4

**Review:**

## Strengths

* **Clear architectural sketch** and motivation; the dual-head design is sensible and compact for an extended abstract.
* Reports **cross-subject transfer** patterns (trained S01, evaluated 01/02/05/07) with consistent ventral/lateral dominance in normalized accuracy.
* Puts **semantic vs. neural objectives** in explicit tension and attempts ROI-level diagnostics (category-wise profiles, MRR vs. voxel r).

## Weaknesses

* **Metric correctness / clipping bug**: "Normalized accuracy" is defined as

    $\text{Acc}_v = \frac{\min(1, r_v^2)}{\max(10^{-8}, \mathrm{NC}_v)}$.

    As written, if NC$_v$<1, Acc$_v$ can exceed 1.0; yet the text states 1.0 denotes ceiling performance. The intended metric should be **$\min\big(1,\, r_v^2/\max(10^{-8},\mathrm{NC}_v)\big)$** (or a bounded alternative such as $r/\sqrt{NC}$). Please fix and recompute all values.
* **Underspecified cross-subject mapping**: The voxel head is ROI-specific linear maps. How are **voxel indices/sizes reconciled across subjects** (different voxel counts per ROI)? Is there a subject-specific head, a shared head with padding, or a second mapping layer? This is critical for the reported cross-subject results.
* **Baselines are missing**: No comparison to (i) **ROI-agnostic linear readout** from frozen DINOv2 tokens, (ii) single-objective trainings (voxel-only vs caption-only), or (iii) contemporary NSD encoders cited in the text. Without these, it’s impossible to attribute gains to the ROI decoder or the dual head.
* **Evidence is thin for the caption head**: The paper names MRR/recall but **reports no retrieval numbers** or plots; divergence claims rely on profiles without concrete metrics or CIs.
* **Training details ambiguous**: "Adam ($10^{-4}$, $10^{-2}$)" is unclear (two LRs for which parts?), missing **weight decay**, **seed control**, **schedule**, **early stopping**, **data splits**, and **subject-wise voxel counts per ROI**.
* **ROI definition**: The choice of **five coarse ROIs** is not justified (how derived from NSD parcellations, boundaries, voxel inclusion criteria?). This matters for interpretability claims.
* **Evaluation reporting**: Table 1 shows only **normalized accuracy** averages per ROI/subject; please add **raw r**, **NC**, and **voxel counts**, with **CIs/SEs** and **voxel-selection criteria** (e.g., reliability thresholding).

## Correctness & Technical Soundness

* **Loss formulation** is standard; however, the **embedding space choice** (MPNet) isn’t motivated vs. CLIP-text or domain-tuned caption encoders; normalization of embeddings before cosine/MSE is not described.
* **Noise ceiling**: Split-half with Spearman–Brown is appropriate, but no details on **number of repeats per image**, split protocol for **shared1k**, or **Monte-Carlo averaging** beyond "100 splits". Please detail.
* **Normalization bug** (above) undermines quantitative conclusions until corrected.

## Evidence & Validation

* The presented **ROI pattern consistency** is interesting, but the absence of **baselines, ablations, uncertainty, and caption metrics** makes the main claims speculative. The extended-abstract format allows limited experiments, but **one quantitative caption table and one ablation** are still expected.

## Reproducibility Signals

* Partial: dataset and broad protocol given; but **code link**, **exact preprocessing**, **ROI construction**, **voxel lists**, **seeds**, **LR schedule**, **weight decay**, and **data split hashes** are missing. Add a minimal **repro appendix or repo**.

## Writing & Presentation

* The abstract and Figure 1 are clear. Table 1 is readable but **insufficient** (needs uncertainty and complementary metrics). Minor ambiguity in training hyperparameters. Overall clean for workshop.

## Questions for Authors (camera-ready guidance)

1. **Fix and re-report the normalized-accuracy metric** with proper clipping: confirm if it’s $\min(1, r^2/NC)$; provide sensitivity vs. alternative bounded forms.
2. **How is cross-subject prediction implemented?** Are voxel heads subject-specific? If shared, how are differing voxel counts per ROI handled? Provide the exact mapping scheme.
3. **Baselines**: (a) ROI-agnostic linear readout from DINOv2 tokens; (b) single-objective trainings (voxel-only, caption-only); (c) caption head with **CLIP-text** to test MPNet choice.
4. **Caption metrics**: Report **MRR/Recall\@k** per ROI and overall, with CIs; include at least one **category-wise numeric table** backing Figure 2’s claim.
5. **ROI definition**: Specify the pipeline mapping NSD parcellations to the five ROIs; report **voxel counts per ROI per subject** and reliability thresholds.
6. **Training details**: State **LR(s)** per module, **weight decay**, **scheduler**, **dropout placement**, **epochs/early stopping**, **seeds**, **hardware**, and **run-time**.

**Score:**

3

**Topic Fit:**

2

---

### Official Review · Reviewer_vW1D · 2025-09-15
**An interesting idea but details and main results are difficult to follow**

**Confidence:** 4

**Review:**

The authors fine-tune a DINO model / add a transformer decoder for dual heads: one for voxel-wise encoding, and the other for ROI-wise caption embedding prediction.

Overall this is an interesting way to look at the role of encoding and semantics in the dual heads. However, the overall motivation and results are somewhat difficult to follow. Specific comments below

(1)  The dual heads are not clearly described in the abstract or introduction. In particular the voxel-wise prediction is described as "regress voxel activations" (abstract) and simply "for voxels" (intro). I believe this head is predicting voxel-wise activity (encoding) but this is very confusing. If that is correct, a more clear wording could be to say that the dual heads are predicting (i) voxel-wise resopnses and (ii) per-ROI caption embeddings.

(2) Contributions are somewhat overstated/repetitive. The dual loss is mentioned in (1) and (2). (3) is simply the training protocol and a worthwhile methodological detail to outline but not a novel contribution.

(3) The findings mention several times that "different locations" provide the peak performance for the captioning task but do not mention what those regions are. Presumably this is shown in 2a but that figure (1) is very difficult to parse and (2) only shows the visual regions mentioned for the voxel-wise encoding task. What are the "different regions" that do a better job of captioning?

These concerns make it difficult to understand the contributions / findings of the paper.

**Score:**

3

**Topic Fit:**

3